Determinants of seasonal changes in availability of food patches for elephants (Loxodonta africana) in a semi-arid African savanna

Clegg Bruce W. bruce@malilangwe.org 1 2
O’Connor Timothy G. 1 3
1 School of Animal, Plant and Environmental Sciences, University of the Witwatersrand , Johannesburg , South Africa
2 The Malilangwe Trust , Chiredzi , Zimbabwe
3 South African Environmental Observation Network (SAEON) , Pretoria , South Africa
Somers Michael
Electronic publication date: 2017 Jun 20
Publication date: 2017
Volume: 5
Electronic Location ID: e3453
Received 2017 Feb 8; Accepted 2017 May 19
Copyright: ©2017 Clegg and O’Connor
Copyright year: 2017
Copyright holder: Clegg and O’Connor
License: This is an open access article distributed under the terms of the Creative Commons Attribution License, which permits unrestricted use, distribution, reproduction and adaptation in any medium and for any purpose provided that it is properly attributed. For attribution, the original author(s), title, publication source (PeerJ) and either DOI or URL of the article must be cited.
License URL: https://creativecommons.org/licenses/by/4.0/

Keywords: Grass, Trees, Shrubs, Bark, Soil, Rainfall, Forbs, Fire

Funding: The Malilangwe Trust This work was funded by The Malilangwe Trust. The funders had no role in study design, data collection and analysis, decision to publish, or preparation of the manuscript.

==============================
Loss of biodiversity caused by impact of elephants (Loxodonta africana) on African woodlands may require a management response, but any action should be based on an understanding of why elephants choose to utilise trees destructively. Comprehension of elephant feeding behaviour requires consideration of the relative value of the plant groups they may potentially consume. Profitability of available food is partly determined by the time to locate a food patch and, therefore, as a foundation for understanding the influence of food availability on diet selection, key controls on the density of grass, forb, and browse patches were investigated across space and time in a semi-arid African savanna. Density of food patches changed seasonally because plant life-forms required different volumes of soil water to produce green forage; and woody plants and forbs responded to long-term changes in soil moisture, while grasses responded to short-term moisture pulses. Soil texture, structure of woody vegetation and fire added further complexity by altering the soil water thresholds required for production of green forage. Interpolating between regularly-timed, ground-based measurements of food density by using modelled soil water as the predictor in regression equations may be a feasible method of quantifying food available to elephants in complex savanna environments.

Introduction

African savannas are characterised by a predominance of woody plants and grasses (Huntley, 1982), but they also support a richness of herbaceous dicotyledons (forbs). Savanna elephants (Loxodonta africana) harvest food from all these plant types (Barnes, 1982; Field, 1971; Kabigumila, 1993), but their conspicuous impact on woody plants has the greatest potential to cause long-term vegetation change (Lamprey et al., 1967; Laws, 1970b; Leuthold, 1977). Over time, this behaviour may simplify the structure and composition of woodlands (O’Connor & Page, 2014), jeopardising the persistence of impacted species (Lombard et al., 2001; O’Connor, Goodman & Clegg, 2007) and the biota that are dependent on the original complexity (Cumming et al., 1997; Herremans, 1995; Kerley & Landman, 2006). Extensive impact on woodlands by elephants was first noticed in the 1930s (Laws, 1970a) and since then many wooded areas in Africa have been converted to shrubland or grassland (Spinage, 1994). The threat of local extirpation of some impacted woody species (O’Connor, Goodman & Clegg, 2007) begs a management response, but any action should be founded upon an understanding of why elephants choose to use woody plants in a destructive manner.

Comprehension of elephant feeding patterns requires consideration of the relative value of the plant groups they may potentially consume. Savanna elephants may utilise grasses, including roots (De Boer et al., 2000; De Longh et al., 2004; Wyatt & Eltringham, 1974), forbs (Barnes, 1982; Field, 1971), and all components of woody plants (Field, 1971; Guy, 1976). They adjust their diet seasonally depending on food availability (Cerling et al., 2004; Owen-Smith, 1988). During the wet summer months large amounts of grass and forbs are eaten, but when these food types dry out in winter, elephants consume an increasing amount of leaves and twigs from woody plants, followed by bark and roots as leaves are shed.

Impact on woody vegetation by elephants is greatest when their feeding involves breaking branches, debarking stems, or toppling, pollarding or uprooting whole plants, and less when trunkloads of leaves are stripped without breaking branches (Clegg, 2010). When diet is composed solely of grass and forbs there is no damage to woody plants. Prediction of diet is therefore a crucial step towards forecasting impact on woody vegetation.

Large mammalian herbivores exercise a hierarchy of decisions, from patch to landscape, when deciding where to forage and what to eat (Senft et al., 1987), which necessitates a landscape-level assessment of the relative availability of forage types. Elephants may forage for 18 h a day to meet their needs (Wyatt & Eltringham, 1974), thus searching time for food patches is critical. Patch density of a forage type therefore provides a functional measure of food availability because the distance between patches determines the searching time to locate food, which is a constraint to intake (Fortin et al., 2015). Food availability therefore differs among vegetation types in relation to differences in food-patch densities. Food availability also varies seasonally over an annual cycle. Accordingly, environmental influences on the leaf phenology of forage types were examined.

Short-term changes in soil water govern grass growth but are apparently less important for leaf flush and drop of woody plants, for which levels of internally stored water (Borchert, 1994; Chapotin, Razanameharizaka & Holbrook, 2006; Kulmatiski et al., 2010), photoperiod and temperature play an important role (Archibald & Scholes, 2007; Choler et al., 2010). However, evidence for an influence of soil water on the phenology of woody plants includes early leaf drop in response to low rainfall in the preceding wet season (Borchert, Rivera & Hagnauer, 2002), woody plants that receive early rain flush first (Borchert, Rivera & Hagnauer, 2002; Clegg, 2010), and that internally stored water has to be recharged from the soil store. Environmental influences on the growth of forbs in African savannas are unstudied.

The amount of soil water that is available for uptake by plants is a function of rainfall pattern, soil storage capacity as influenced by profile depth and soil texture, and evapo-transpirative demand (Ritchie, 1981). Clay-rich soils store more water at field capacity than sandy soils but this is partly offset by clay soils holding a greater proportion of water more tightly within the pores (Foth, 1990). A tendency for higher water-use efficiency (WUE) of tropical C4 grasses relative to tropical C3 forbs and woody plants (Ehleringer & Monson, 1993) may lead to differences in leaf phenology between these plant groups. Inter-plant competition for soil moisture influences the duration of leaf carriage in savanna (Smit, 2001).

Available nutrients and fire also influence food availability. The influence of soil nutrients corresponds directly with that of soil texture (Foth, 1990), an effect that cannot be isolated from the influence of soil texture on soil water availability. That fire has an immediate effect on food availability is obvious, but subsequent effects on plant phenology and available forage are poorly understood.

The aim of this study was to investigate key determinants of food available to elephants across space (vegetation types) and time (annual cycle) as a foundation for understanding forage selection. The following specific questions were addressed: (1) Can plant-available water in the soil be used to predict the availability of the main food types; (2) Do different food types differ in their annual pattern of availability; (3) Is the relation between food availability and soil water influenced by soil texture; (4) Does soil texture influence the effect of fire on food availability; (5) Does the structure of woody vegetation influence the annual phenological pattern?

The study formed the foundation of a landscape scale analysis of elephant foraging behaviour that will be reported elsewhere. The findings are relevant to savanna elephants throughout Africa, and are also applicable to other herbivores that inhabit savannas globally. The results concerning the relationship between soil water and the leaf phenology of the plant growth-forms are novel and, although particular emphasis was given to foraging elephants, the findings also make a contribution to the general theory of savanna function.

Materials & Methods

The study was conducted in the semi-arid savanna of Malilangwe Wildlife Reserve in south-eastern Zimbabwe (20°58′–21°15′S, 31°47′–32°01′E). The reserve has a hot wet season from November to March, a cool dry season from March to August, and a hot dry season from September to October. Mean annual rainfall is 557 mm (n = 64; CV = 34.2%), with approximately 84% falling in the hot wet season. Rainfall during the year under study was 716 mm. The average minimum and maximum monthly temperatures range from 13.4 °C (July) to 23.7 °C (December), and 23.2 °C (June) to 33.9 °C (November) respectively (Clegg, 2010). Frost is rare. Thirty-eight vegetation types, from open grassland to dry deciduous forest, have been identified on seven geological types, with soils ranging from 90% sand to 41% clay (Clegg & O’Connor, 2012). Fire has been used as a tool for rangeland management since 1994. Management burns were conducted in October 2001, five months prior to the start of the study period. Permission to conduct the study was granted by the Director of The Malilangwe Trust.

The study focused on grass, forbs, and leaves and bark from woody plants because these constitute the bulk of an elephant’s diet (Barnes, 1982; De Boer et al., 2000; Field, 1971; Guy, 1976). Maps of the areas burnt during the previous year and areas that experienced early woody leaf flush (identified from an October 2002 landsat 7 ETM+ image using a Normalized Vegetation Index) were combined with a fine-scale vegetation map (Clegg & O’Connor, 2012) to create a spatial framework for sampling with 65 landscape units. Within each unit, the relationship between the density of grass, forb, or leaf patches and soil water was investigated between March 2002 and March 2003. The density of bark patches, whose availability was considered constant for a year, was estimated for each landscape unit. Sampling was undertaken between November 2001 and July 2003 at approximately three-month intervals. On each occasion, five sample points were positioned in each landscape unit using a stratified random strategy and located in the field using a GPS. Plant nomenclature follows Mapaura & Timberlake (2004).

Estimation of the density of food patches

Grass and forbs

An individual grass tuft or forb plant would constitute a patch if it was growing in isolation, but at high plant densities a trunkload would be made up of material from multiple plants. A 1 m2 area was therefore arbitrarily defined as the patch size for grasses and forbs.

The nutritional quality of savanna grasses varies over the annual cycle (Buxton & Redfearn, 1997; Lyons, Machen & Forbes, 1996). Elephants avoid eating senescent forage, so a plant can only be considered a patch if it offers a sufficient ratio of green to dry material (Clegg, 2010; Loarie, Van Aarde & Pimm, 2009). Accordingly, the density of patches with green grass only, or a mixture of green and dry grass were estimated. In support of recognising two grass patch types, elephants require twice the time to harvest, clean by shaking, chew and ingest a trunkload of mixed grass than one of green grass (Clegg & O’Connor, 2016). When most forbs senesce their leaves are abscised and their stems wither, therefore only the density of patches with green forbs was estimated.

At each sampling point, 25 1 m2 quadrats were sampled along a 50 m tape. The cover of green grass, dry grass and green forbs was estimated in each quadrat using an eight point scale (0%; 1%; 2–10%; 11–25%; 26–50%; 51–75%; 76–95%; 96–100%). A quadrat was assigned to a patch type using the following criteria: green grass = green grass cover ≥1%, ≥90% of total grass biomass green, and total grass cover ≥1%; mixed green and dry grass ≤90% and ≥10% of total grass biomass green; green forb = green forb cover ≥2%. The density of patches (m−2) with green grass, mixed green and dry grass, and green forbs was calculated per site.

Leaves from woody plants

Individual shrubs and trees were considered patches. Elephants avoid eating senescent leaves (Clegg, 2010) and for this reason only woody plants with >25% of their canopies with green leaf were used to estimate patch density. Below this level it becomes difficult for elephants to harvest a trunkload that is comprised solely of green leaves (Clegg, 2010). Elephants also avoid very young, red leaves of Colophospermum mopane possibly because of a high concentration of polyphenols and resins (Styles & Skinner, 1997), and therefore mopane plants with this leaf type were excluded. Other species that were also excluded because they were avoided by elephants were Courbonia glauca, Euclea divinorum, Salvadora persica, and Thilachium africanum.

The density of woody plants with green leaf was estimated in the following way. At each site the nearest woody plant was assessed and the following recorded: (1) species, (2) shrub (≤3 m) or tree (>3 m), (3) per cent of canopy volume with new leaves, and (4) per cent of canopy volume with mature green leaves, using the eight-point scale. Then the next nearest individual was assessed in the same way. This process was repeated until five trees and five shrubs of each species had been sampled. To reduce sampling, only species with a canopy volume >25 and >15 m3 ha−1 were sampled for trees and shrubs respectively (calculation of volume given in Clegg & O’Connor (2012)). For each species of tree and shrub, an average (n = 25) estimate was derived for each date. To improve the temporal resolution for each shrub and tree species, data were pooled across landscape units with similar topo-edaphic conditions. The per cent canopy volume with new green leaf (Vnew) and mature green leaf (Vmature) was estimated, on a daily basis, for each species of shrub and tree in a landscape unit by interpolating between data points using a smoothing spline regression (KyensLab, 2002). The total per cent canopy volume with green leaf was calculated, for each shrub and tree species in a landscape unit, as: Vtotal=Vnew+Vmature.

Bull elephants are taller than cows and are therefore capable of stripping leaves from a greater height. It was determined from field measurements that bulls and cows (n = 200 each) can feed from a maximum height of just over 6 and 4 m, respectively (see Clegg, 2010 for details). The density of woody plants with green leaf available to bulls (Dbull) was calculated on a daily basis, for each vegetation unit, as: Dbull= ∑i=1ndi,tree,bullpi,tree+ ∑i=1ndi,shrubpi,shrub,

where di,tree,bull is the density of the ith tree species with canopy volume below 6 m (see Clegg & O’Connor (2012) for how density of each species was calculated), pi,tree is the Boolean probability (0 or 1) that Vtotal,i,tree >25%, di,shrub is the density of the ith shrub species and pi,shrub is the Boolean probability (0 or 1) that Vtotal,i,shrub >25%. The calculation was the same for cows except di,tree,cow was based on trees and shrubs with canopy volume below 4 m.

Bark

Individual shrubs and trees of species whose branches were commonly chewed for bark were considered patches. The density of patches with bark available to bulls, in each landscape unit, was calculated as: Dbull= ∑i=1ndi,shrub+ ∑i=1ndi,tree,bull,

where di,shrub is the density of the ith shrub species utilised for bark and di,tree,bull is the density of the ith tree species that was utilised for bark with canopy below 6 m. The calculation was the same for cows except di,tree was based on the density of trees with canopy below 4 m.

Estimation of soil water

A simple model of soil water that estimated water in the upper soil layer and ignored the process of deep drainage was considered adequate. Daily moisture in the top 30 cm of soil was extracted for each sample site from maps generated by linking a model of soil moisture balance (Hobbs, Sparrow & Landsberg, 1994) to a geographic information system. A constant profile depth of 30 cm was used because 50% of the study area had soil ≤30 cm deep (Clegg & O’Connor, 2012) and in savannas most active roots of woody and herbaceous plants occur within this depth (Kulmatiski et al., 2010). Daily moisture loss from the 0–30 cm profile was modelled using a negative exponential function that was dependent on available soil moisture and driven by daily rainfall and potential evaporation: Mt=Mt−1+Rt×exp−k.PE,

where M is the soil moisture fraction, R is the rainfall fraction, PE is potential evaporation (mm) and k is an evaporative constant. M was scaled between 0 and 1, and was calculated as: M=msoil−mminmFC−mmin,

where msoil is the moisture (mm) in the 0–30 cm profile, mmin is the minimum air dried moisture (mm), and mFC is the field capacity of the 0–30 cm profile. If the soil profile was less than 30 cm, mFC was calculated for the actual soil depth. Minimum air dried moisture was calculated for each vegetation unit from per cent silt and clay (Clegg & O’Connor, 2012) using the equation of Bennie et al. (1988). Field capacity was calculated using the equation of Hutson (1984). R was scaled between 0 and 1 and was calculated by dividing daily rainfall (mm) by mFC − mmin. Rainfall data were collected from 14 gauges within and 12 gauges outside the study area. A rainfall surface was generated for each rainfall event by interpolating between gauges using a triangular irregular network (Eastman, 2003). Following Hobbs, Sparrow & Landsberg (1994) the evaporative constant k was derived from mmin. Daily PE data were collected from two class A evaporation pans that were located at the Zimbabwe Sugar Association, 27 km to the west of Malilangwe headquarters.

Relationship between density of food patches and soil water

Curve fitting

For each landscape unit, the average density of food patches (D) on a particular date was plotted against average soil moisture for the previous 5, 10, 15, 20, 25, 30, 45, 60, 75, 90, 105, 120, 135 and 150 days. The relationship for green grass and forbs could be described by a monotonically increasing sigmoidal curve, with lower and upper asymptotes 0 and 1 respectively (Fig. 1). Green leaves from woody plants showed a similar relationship, but the data for each land unit required standardisation to achieve asymptotes of 0 and 1. Consequently, for these forage types the logistic equation: D=11+expa−bM,

where a and b are constants, was fitted to the data from each land unit using Systat 9 (SPSS, Chicago, IL, USA).

Figure 1 Sigmoidal curves representing the relationship between availability and soil water for the forage types.

M1, threshold between lag and exponential phases (green-up); M2, threshold between exponential and plateau phases (plateau); M3, threshold between plateau and exponential phases of the decline in mixed grass patches and M4, threshold between exponential and lag phases of the decline in mixed grass patches.

The relationship between the density of mixed grass patches and soil moisture was initially a monotonically increasing sigmoidal curve that was followed at higher levels of soil moisture by a monotonically decreasing sigmoidal curve (Fig. 1). Consequently, the following model was fitted to the data for density of mixed grass patches: D=11+expa−bM×11+expc−dM,

where a, b, c and d are constants.

For each landscape unit, the curve from the predictor variable with the best fit was chosen to represent the relationship between the density of the food type and soil water.

Soil water thresholds

The sigmoidal relationship between the density of forage patches and soil water can be divided into lag, exponential and plateau phases. The soil water thresholds marking the start and end points of these phases can be represented by the points of maximum curvature on the sigmoidal curves (Fig. 1). The first threshold (M1) represents the point at which availability begins to increase rapidly (green-up) and the second (M2) the point when availability nears its maximum (plateau). Mixed grass had two additional thresholds, M3 and M4, for the declining phase of its relationship with soil water when mixed grass is converted into green grass. For each landscape unit, these thresholds were determined for the food types by calculating the local maxima of curvature.

Influence of soil texture, plant life-form and fire on food availability

Moderated multiple regression (Aguinis, 2004) was used to determine whether the linear relationship between soil water threshold and soil clay content was dependent on the value of a dichotomous moderator variable, which was either dummy coded plant life-form (e.g., green grass = 1, green forbs = 0) or dummy coded fire (burnt = 1, unburnt = 0). Separate analyses were conducted for green-up and plateau thresholds and for each paired combination of plant life-form. We used the standard method of determining whether a moderating effect existed, which entailed the addition of a linear interaction term in the multiple regression model, Y=b0+b1X+b2Z+b3XZ,

where Y is the soil water threshold (mm), X is soil clay content (%) and Z is the dichotomous moderator variable (either plant life-form or fire). A significant interaction term (b3) indicates that the association between X and Y varied as a function of Z, but does not specify the form of the interaction. Consequently, a significant (P < 0.05) interaction was probed further by using the Johnson-Neyman technique (Kowalski, Schneiderman & Willis, 1994) to calculate the regions along the textural gradient where the moderating factor had a significant effect. The regions of significance (RoS) were calculated using the online application: Probing Interactions in Moderated Multiple Regression and Differential Susceptibility Research (http://www.yourpersonality.net/interaction/) that was designed by R. C. Fraley as a supplement to Roisman et al. (2012).

Tests of the assumptions of moderated regression were conducted following Lund & Lund (2013). Outliers (deleted studentized residuals >±2.0) were removed only if there was an obvious ecological reason (e.g., soil water estimates were inaccurate because of a steep slope or proximity to a perennial river), and if removal had a significant effect on the regression results.

Effect of soil texture and structure of woody vegetation on retention of green leaf patches

The maximum density of woody green leaf patches (when all woody plant canopies have >25% green leaf) for each landscape unit was determined. The predictable seasonal decline in patch density due to leaf drop as the dry season progresses and increase with leaf flush following commencement of rains allowed per cent patch retention to be calculated for each unburnt landscape unit by dividing the sum of daily patch density estimates by the maximum possible patch density over the annual cycle. The relationship between per cent patch retention, soil texture and woody canopy volume was then explored using the multiple non-linear regression routine of Labfit (Silva & Silva, 2011). Labfit searches a library of 280 functions and outputs the one that best fits the data. Six landscape units were removed from the analysis because they were affected by water-related environmental anomalies.

Figure 2 Effect of soil clay content and lag period on the temporal pattern of soil water.

Results

Temporal pattern of soil water

Soil water increased sharply when it rained and declined more slowly as soil dried (Fig. 2). As expected, soil water was highest in the wet season and clay soils stored more water per unit depth than sandy ones. Time series constructed using short time lags had large, frequent fluctuations, while longer lags resulted in smoother profiles.

Relationship between density of food patches and soil water

Density of herbaceous forage patches was successfully modelled from soil water using the sigmoidal functions (Fig. 3). Average Radj2 values (±SD) were 0.92 ± 0.14, 0.90 ± 0.11, and 0.89 ± 0.11 for green grass, mixed grass and green forbs respectively. Density of woody plants with green leaves was successfully modelled (average Radj2=0.90±0.08) during leaf drop and flush (Fig. 3) but not during the period prior to leaf drop because at this time woody plants had more green leaf than predicted by soil water (Fig. 4).

Figure 3 Examples of the relationship between availability of food types and soil water in relation to increasing clay content.

Mixed grass (A); green grass (B); green forbs (C); green leaves (D). Data for green leaves from woody plants are restricted to the period from the start of leaf drop, through leaf flush, to maximum patch density.

Figure 4 Examples of temporal estimates of availability of woody green leaf available to adult bull elephants from field measurements and modelling along a gradient of increasing soil clay content.

Measured and modelled estimates corresponded well except for the period before leaf drop.

Plant life-forms responded to changes in soil water over different time scales. Grass responded to short-term fluctuations, with a best predictor average (±95% confidence interval) of 29.7 ± 5.5 and 29.9 ± 5.6 days for green and mixed grass respectively. Forbsresponded to medium-term fluctuations, with a best predictor average of 40.6 ± 5.9 days, and leaves from woody plants to long-term fluctuations, with an average of 99.2 ± 8.0 days.

Influence of soil texture and food type on soil water thresholds

For all food types, the volume of soil water required to start green-up and reach maximum patch density increased linearly with clay content (P < 0.01) (Fig. 5). Thresholds for woody green leaves and mixed grass were well below the field capacity of the soil, the gap between field capacity and threshold increasing with clay content. In contrast, plateau thresholds for green grass were almost at field capacity and were above field capacity for green forbs on clay-rich substrates, indicating that the assumed plateau density of 1 patch m−2 was too high for green forbs.

Figure 5 Linear regressions between clay content and field capacity of the 30 cm soil profile and green-up (M1) and plateau (M2) soil water thresholds for each food type.

Green leaves (A); mixed grass (B); green grass (C); green forbs (D). Equations, adjusted R2 and P values for regressions: green leaves, M1 = 11.022 + 0.563X(R2 = 0.81, P < 0.001), M2 = 12.645 + 0.663X(R2 = 0.87, P < 0.001); mixed grass, M1 = 8.346 + 0.724X(R2 = 0.64, P < 0.001), M2 = 13.423 + 0.774X(R2 = 0.54, P < 0.001); green grass, M1 = 18.671 + 1.225X(R2 = 0.69, P < 0.001), M2 = 26.22 + 1.673(R2 = 0.70, P < 0.001); green forbs, M1 = 10.52 + 1.502X(R2 = 0.68, P < 0.001), M2 = 18.493 + 2.63X(R2 = 0.68, P < 0.001).

Woody plants required significantly (P < 0.05) less soil water to start greening up and reach maximum patch density than green grass and forbs (Tables 1 and 2, Figs. 6 and 7). Green grass and forbs had the same green-up thresholds (P > 0.05), but plateau thresholds were higher for forbs on soils with greater than 14.2% clay. Thresholds for mixed grass were generally similar to woody plants, but differences occurred on some soil types.

Table 1 Regression estimates and region of significance limits (RoS X) for the influence of soil texture and plant life-form on soil water thresholds.

	Dummy coded Moderator variable (Z)	Regression estimates	RoS X (Clay %)	
Threshold (Y)	1	0	b0	b1	b2	b3	XZΔR2	p	Lower bound	Upper bound	
M1	G. leaves	M. grass	8.164**	0.724**	3.342*	−0.174*	0.015	0.019	32.8	41.5	
M1	G. leaves	G. grass	18.671**	1.225**	−7.650**	−0.662**	0.06	0.0005	4.0	41.5	
M1	G. leaves	G. forbs	14.543**	1.458**	−5.682*	−0.783**	0.087	0.0005	4.0	41.5	
M1	M. grass	G. grass	18.671**	1.225**	−10.325	−0.501**	0.027	0.0005	4.0	41.5	
M1	M. grass	G. forbs	10.518**	1.502**	−2.172	−0.779**	0.058	0.0005	5.6	41.5	
M1	G. grass	G. forbs	10.518**	1.502**	8.153	−0.278	0.007	0.22	–	–	
M2	G. leaves	M. grass	12.129**	0.82	1.469	−0.226*	0.02	0.016	15.5	41.5	
M2	G. leaves	G. grass	25.984**	1.674**	−12.386**	−1.080**	0.075	0.0005	4.0	41.5	
M2	G. leaves	G. forbs	21.025**	2.475**	−7.427	−1.882**	0.111	0.0005	4.0	41.5	
M2	M. grass	G. grass	25.984**	1.674**	−12.883**	−0.866**	0.046	0.0005	4.0	41.5	
M2	M. grass	G. forbs	21.025**	2.475**	−7.924	−1.667**	0.084	0.0005	4.0	41.5	
M2	G. grass	G. forbs	21.374**	2.529**	4.610	−0.855*	0.025	0.018	14.2	41.5	
Notes.

* P < 0.05.

** P < 0.01.

Table 2 Regression estimates and region of significance limits (RoS X) for the influence of soil texture and fire (>5 months previously) on soil water thresholds.

	Dummy coded moderator variable (Z)	Regression estimates	RoS X (Clay %)	
Threshold (Y)	1	0	b0	b1	b2	b3	XZΔR2	p	Lower bound	Upper bound	
G. leaves M1	Burnt	Unburnt	11.134**	0.581**	0.498	−0.077	0.003	0.202	–	–	
M. grass M1	Burnt	Unburnt	8.346**	0.724**	6.861*	−0.301	0.026	0.1	–	–	
G. grass M1	Burnt	Unburnt	15.298**	1.461**	7.626	−0.536*	0.029	0.022	23.6	41.5	
G. forbs M1	Burnt	Unburnt	10.518**	1.502**	12.336	−0.862*	0.052	0.012	23.4	41.5	
G. leaves M2	Burnt	Unburnt	13.144**	0.631**	−0.776	−0.059	0.002	0.403	–	–	
M. grass M2	Burnt	Unburnt	12.129**	0.820**	5.742	−0.277	0.018	0.130	–	–	
G. grass M2	Burnt	unburnt	22.807**	1.973**	−0.463	−0.629*	0.021	0.040	12.5	41.5	
G. forbs M2	Burnt	unburnt	21.025**	2.475**	11.389	−1.381*	0.046	0.025	18.5	41.5	
M. grass M3	Burnt	Unburnt	19.349**	1.285**	7.228	−0.627*	0.041	0.016	20.7	41.5	
M. grass M4	Burnt	Unburnt	23.848**	1.983**	7.985	−1.087**	0.053	0.006	15.9	41.5	
Notes.

* P < 0.05.

** P < 0.01.

Figure 6 Effect of soil clay content and food type on green-up (M1) soil water thresholds.

Regions of significant (P < 0.05) differences between green leaves and mixed grass (A), green leaves and green grass (B), green leaves and green forbs (C), mixed grass and green grass (D), mixed grass and green forbs (E), and green grass and green forbs (F) are shaded grey.

Figure 7 Effect of soil clay content and food type on plateau (M2) soil water thresholds.

Regions of significant (P < 0.05) differences between green leaves and mixed grass (A), green leaves and green grass (B), green leaves and green forbs (C), mixed grass and green grass (D), mixed grass and green forbs (E), and green grass and green forbs (F) are shaded grey.

Temporal pattern of food availability

Availability of green and mixed grass was characterised by large, short-term fluctuations (Fig. 8). In contrast, green forbs and woody green leaves had smooth temporal profiles, giving the impression that availability was unrelated to soil water. Mixed grass had lower green-up and plateau thresholds than green grass and was therefore more consistently available over the annual cycle. Green grass only became plentiful for a short period during the late rainy season when its thresholds were exceeded.

Figure 8 Temporal availability of forage types along a gradient of increasing clay content.

Data for woody green leaves are for adult bulls. Availability of woody green leaf was slightly lower for cows, but otherwise followed the same pattern.

Maximum patch density was higher for herbaceous food types than for woody green leaves and bark. The plateau threshold was not reached for forbs. Food was less available on clay-rich substrates than on sandy ones, but considering the higher growth thresholds on clay soils, the difference was lower than expected.

Effect of fire on food availability

The relationship between soil water and density of woody green leaf patches was not affected by fire, but green-up and plateau thresholds of green grass and green forbs, and the M3 and M4 thresholds of mixed grass were affected (Table 2). Fire lowered (P < 0.05) the thresholds for these food types on clay-rich soils but not on sandy ones (Fig. 9). This meant that on clay-rich substrates, density of green grass or green forb patches was higher on burnt than unburnt areas for a given volume of soil water, and that mixed grass was converted to green grass at lower levels of soil water on burnt compared to unburnt sites (Fig. 10).

Effect of soil texture and woody vegetation structure on retention of green leaf patches

The effect of soil clay content and woody canopy volume on retention of green leaf patches was best represented (P<0.005,Radj2=0.379) by the power function Y=a∗X1X2b,

where Y is per cent patch retention, X1 is per cent clay content of the top 30 cm of soil, X2 is woody canopy volume (m3 ha−1) and a and b are constants. Estimates of a and b were 106.9 and 0.059 respectively. Patch retention increased as clay content increased and woody canopy volume decreased (Fig. 11).

Figure 9 Effect of a significant interaction between soil clay content and fire (>5 months previously) on soil water thresholds for green grass, green forbs and mixed grass.

Green grass M1 (A); green grass M2 (B); green forbs M1 (C); green forbs M2 (D); mixed grass M3 (E); mixed grass M4 (F). Regions of significant (P < 0.05) difference between burnt and unburnt areas are shaded grey.

Figure 10 Examples of the effect of fire (>5 months previously) on the patch density—soil water relationship for herbaceous forage types along a gradient of increasing clay content.

Figure 11 Influence of soil clay content and woody canopy volume on retention of green leaf patches.

Y = a∗(X1∕X2)b, where Y = patch retention, X1 = soil clay content, X2 = woody canopy volume, and a and b are constants.

Discussion

Determinants of seasonal changes in food availability

Savannas have been intensively studied (Sankaran et al., 2005), but drivers of forage production in these environments are still only superficially understood. In this study, food availability for elephants changed seasonally primarily because grasses, forbs and woody plants required different volumes of soil water for development and maintenance of green foliage, and they responded to fluctuations in soil moisture over different time scales. Soil texture, the structure of woody vegetation and fire added further complexity by altering the soil water thresholds required by the plant groups for production of green forage. Differences were apparent among growth forms despite each plant group being comprised of a large number of species (woody plants = 89, grasses = 66, forbs = 72).

Woody plants required less water per unit volume of soil than grasses and forbs to produce green foliage. This is possibly because they have more laterally extensive root systems that harvest water over a wider area, some species are deeply-rooted allowing access to deep-stored water, or because they have access to an additional source of water stored in their wood that can be drawn upon when soil is dry (Borchert, 1994; Chapotin, Razanameharizaka & Holbrook, 2006; Kulmatiski et al., 2010).

Grasses responded to short-term fluctuations in soil water, while woody plants and forbs responded to longer seasonal changes. Grasses appear to be geared for exploiting moisture pulses even if they are short-lived (<1 month) and unseasonal, while woody plants and forbs are more conservative, only responding to the more predictable, longer-term (1–3 month) seasonal moisture cycles (Archibald & Scholes, 2007). This difference may be explained by the ability of woody plants to store water internally.

Soil texture in combination with plant life-form added further complexity to the relationship between food availability and soil water. For all food types, green-up and plateau soil water thresholds increased linearly with increasing clay content. This was expected because although clay soils store more water per unit volume than sandy soils, water is more difficult to extract from clay soils (Foth, 1990). Despite this, woody plants retained leaf for longer on clay-rich than on sandy soils. This suggests that the effect of soil texture on food availability is determined by the difference between field capacity and soil water threshold rather than the magnitude of the soil water threshold alone. Density of forage patches will remain at maximum until soil water drops below the plateau threshold, and how long this takes to happen is determined by the volume of water that the soil can store over and above the plateau threshold (field capacity - plateau soil water threshold). For woody green leaves and mixed grass, this volume is relatively large on sandy soils and becomes even larger with increasing clay content (Fig. 7). For green grass and green forbs, however, the volume stored in excess of the plateau threshold is relatively small on sandy soils and becomes even smaller with increasing clay content. The net result is that leaves on woody plants and mixed grass stay green for longer on clay-rich substrates compared to sandy ones, with the reverse being true for green grass and green forbs. However, this will only happen if rainfall is sufficient to fill the clay soils to capacity, as was the case in this study.

On sandy soils forbs required the same volume of soil water as grass to reach maximum patch density, but required more water than grass on clay-rich substrates. A possible explanation is that greater water use efficiency of C4 photosynthesis affords grasses an advantage over C3 forbs on clay-rich substrates, where soil water is more difficult to extract, but not on sandy soils where water is held less tightly (Ehleringer & Monson, 1993).

Woody vegetation structure in combination with soil texture further influenced the availability of green browse. Shrubs and trees retained green leaves for longer when the canopy volume of woody plants was low, which was generally the case on clay-rich substrates, and shed leaves earlier when canopy volume was high, which was often the case on sandy soils. In accordance, mopane trees carried leaves for longer into the dry season following tree thinning (Smit, 2001), ostensibly as a result of reduced inter-tree competition when trees are widely spaced (Smith & Goodman, 1986; Walker & Smith, 1983). The net result of woody vegetation structure was a high maximum density of browse patches on sandy and loam soils that declined to low density early in the dry season; and a low maximum density of browse patches on clay-rich substrates that persisted for long into the dry season. Clay-rich substrates may therefore support a higher density of browse patches than sandy soils during the mid-dry season, despite a lower density of woody plants.

Fire influenced the temporal availability of food for elephants, but only for herbaceous vegetation growing on clay-rich substrates. The study was conducted during the second growing season after fire when the immediate effects of fire, such as barren, blackened soil surfaces, had already been ameliorated by regrowth of herbaceous vegetation during the 2001/2002 growing season. For a given volume of soil water, density of mixed grass, green grass, and green forb patches was higher at burnt than unburnt sites. At Malilangwe, clay soils support greater grass biomass than sandy ones (Clegg, 1999) and therefore greater competition among grass plants for soil water is expected on clay-rich substrates. In savannas, biomass of grass may be reduced for several years post-fire (Frost & Robertson, 1985; Savadogo et al., 2009; Snyman, 2005), with the extent of reduction determined by fire temperature (Frost & Robertson, 1985). At Malilangwe, hotter fires because of larger fuel loads, and therefore greater post-fire reductions in grass biomass are expected on clay compared to sandy substrates. This may result in the available soil water being spread over less herbaceous biomass post-fire on clay-rich soils, and hence the lower green-up and plateau soil water thresholds at burnt sites. In addition, because senescent material carried over from previous seasons is removed by fire, grass has a higher ratio of live-to-dead material at burnt compared to unburnt sites (Shombe et al., 2008; Van de Vijver, Poot & Prins, 1999), which further explains the higher density of green herbaceous patches in post-fire regrowth. Plants defoliated by fire may also have higher root-to-shoot ratios (Snyman, 2005), and a larger number of secondary and fine tertiary roots (Hartnett, Potgieter & Wilson, 2005) than unburnt plants which could enable them to rehydrate and green-up using less water (Fisher, 1978).

Although fire reduced the canopy volume of the shrub layer, it had little effect on total woody canopy volume because the tree layer was mostly out of reach of the flames. Consequently fire did not reduce inter-tree competition for soil water, which may explain why there was no post-burn effect on the temporal availability of green browse.

Although findings concerning the relationship between soil water, soil texture, fire, structure of the woody layer and leaf phenology have been presented in the context of food for elephants, they also have obvious significance for savannas in general. Specifically, the results support the hypothesis that soil water plays a central role in the tree-grass interaction (Walter, 1939; Walter, 1970) and that explanation of this role is a key challenge towards understanding the functioning of savannas. Furthermore, we believe the insight gained into the phenological pattern of forbs suggests that the hitherto reported tree-grass interaction of savannas should be expanded to a tree-grass-forb interaction. However, the study was not perfectly designed to address these issues so we have purposely chosen not to elaborate further on these points.

Implications for foraging elephants

In savannas elephants are confronted with a broad spectrum of food choices that change across space and time. Change over the annual cycle is predictable because there is a progressive drying of soil from the onset of the dry season, with food types being lost from the choice set in a sequence determined by their relative soil water requirements. Food types with the highest requirement are lost first, followed by the next highest and so on until only the most drought resistant types remain. This corresponds with reported seasonal changes in the diet of elephants from consumption of forbs and grass (high water requirement) in the rainy season to mostly leaves from woody plants (medium water requirement) when rains end, followed by an increasing amount of bark and roots (low water requirement) in the late dry season (Barnes, 1982; Cerling et al., 2004; Field, 1971; Guy, 1976). This understanding enables prediction of diet under different scenarios. For example, in semi-arid savannas grass is expected to be utilised for longer in wet than in dry years, and consumption of bark is expected to increase during drought.

Knowledge of the mechanisms that cause spatial variation in food availability can help to explain seasonal movements of elephants. For example, because trees and grasses have different soil water requirements, key browse and grass resources are often separated geographically and elephants must undertake seasonal movements between them if they are to exploit their different phenological patterns. This may partly explain why elephants are often reported to move between a wet season range, where grass makes up the bulk of the diet, and a dry season range where browse from woody plants is the dominant food type (Cerling et al., 2006; Field, 1971; Loarie, Van Aarde & Pimm, 2009). To locate and utilise patches of green grass elephants inhabiting savanna environments must track rain storms that are typically isolated and scattered (Eisinger & Wiegand, 2008), which forces them to move fast (Birkett et al., 2012) and widely during the rainy season (Bohrer et al., 2014; Thouless, 1995; Young, Ferreira & Aarde, 2009). In contrast, browse from woody plants is a less ephemeral source of food and therefore elephants can move more slowly (Birkett et al., 2012) and occupy a relatively confined area during the dry season (Bohrer et al., 2014; Thouless, 1995; Young, Ferreira & Aarde, 2009).

A key challenge to the study of feeding behaviour of a wide-ranging herbivore in a complex environment is to quantify available food at an adequate spatial and temporal resolution. The approach described in this study may meet this need. The daily food supply for elephants at Malilangwe was successfully mapped for one annual cycle using this method (Clegg, 2010), and the results successfully used to interpret foraging decisions that yielded generalised insight into elephant feeding behaviour that has Africa wide relevance (for monthly maps showing the spatial availability of the different forage types over the study area see Clegg, 2010 pgs 60, 61, 66, 72–74). Although time consuming, a ground-based approach was identified as a key to this success because estimates of patch density and the vertical distribution of browse on a species-specific basis could be derived for each food type, which is currently not possible with existing satellite technology. These measures were essential for estimating the mass and energy content of trunkloads harvested from food patches (Clegg, 2010).

The findings of this study also have relevance for other large mammalian herbivores that inhabit savanna environments. However, when extrapolating the results it should be recognised that elephant-centric nuances were embedded in the approach and methodology of this study. For example, availability of browse patches was calculated using a maximum reach height of 4 m and 6 m for cows and bulls respectively, and browse patches were only considered available if they had >25% green leaf. These nuances make the results specific to elephants. However, despite this, the discerning reader should still be able to recognise the implications for other herbivores.

Conclusions

The types of food available to elephants changed seasonally because (1) plant life-forms required different volumes of soil water to produce green forage; (2) woody plants and forbs responded to long-term changes in soil moisture, while grasses responded to short-term moisture pulses; and (3) soil texture, the structure of woody vegetation and fire added further complexity to the pattern of food availability by altering the soil water thresholds required by the plant groups for the production of green foliage.

Interpolating between regularly-timed, ground-based measurements of food density by using modelled soil water as the predictor in regression equations proved to be a feasible method of for quantifying the food available to elephants over an annual cycle. In complex environments, stratification on the basis of soil texture, woody vegetation structure and recent fire history is essential.

Supplemental Information

Supplemental Information 1 Raw data for temporal change in patch density for each plant life form

Click here for additional data file.

Supplemental Information 2 Raw data for the relationship between patch density and soil water for each plant life form

Click here for additional data file.

The authors thank Julius Matsuve, Julius Shimbani, Cryson Chinondo and Philmon Chivambu for assisting with data collection in the field. Special thanks are given to Keith Clegg for his help with constructing the figures.

Additional Information and Declarations

Competing Interests

Author Contributions

Data Availability

The authors declare there are no competing interests.

Bruce W. Clegg conceived and designed the experiments, performed the experiments, analyzed the data, contributed reagents/materials/analysis tools, wrote the paper, prepared figures and/or tables, reviewed drafts of the paper.

Timothy G. O’Connor conceived and designed the experiments, wrote the paper, reviewed drafts of the paper.

The following information was supplied regarding data availability:

The raw data has been supplied as a Supplementary File.

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
