# Peer review of "Determinants of seasonal changes in availability of food patches for elephants (Loxodonta africana) in a semi-arid African savanna"

_PeerJ, doi:10.7717/peerj.3453_

## Round 0.1 · original submission · Minor Revisions

One referee suggests some form of presentation of the spatial variation in responses. The other suggests focusing more on the dynamics forage resources, and to discuss the broader implications (see comments below).

Reviewer 1 ·

Basic reporting

The authors state that the actual modeling of the foraging responses of elephants to the dynamics of forage availability presented in this paper will be published elsewhere. Therefore the investigation is of interest for all herbivores, not just elephants, and the results are relevant for semi-arid savannas functioning with herbivore disturbances. I believe what the paper shows is that by finely describing processes behind forage dynamics (availability and quality) one can model the herbivore foraging behavior and predict animal movement. Here elephant is a species that adequately tests this approach (but not shown). I do not see why elephant and their impact on savannas have to be put at the fore-front in this paper, there is little discussion about that afterwards, it could shorten the introduction.

Experimental design

No comment

Validity of the findings

No comment

Additional comments

Great field work. A good read. I feel the paper could be shortened slightly, but broadened in scope,by focusing on the dynamics forage resources, and discuss the implications for elephants as well as other herbivores. I understand that the present study was triggered by an elephant study, but the level of information gathered is well beyond elephant foraging.

·

Basic reporting

The paper is well written. The literature review is sufficient, but suggest that the authors also access the recent literature on nutrient influences on elephant foraging decisions by Pretorius et al and Kohi et al. Also, Wooley et al demonstrate nutrient basis for elephant diet switching. The data are comprehensively reported, but this results in a large number of figures and tables, some of which could be moved to supplementary material to make the final paper more readable.

Experimental design

The paper is primary original research. The search questions Are well defined. The paper provides a robust emchanistic understanding of heterogeneity in savanna system looking at water as a primary driver, and this is important basis for explaining animal behavior, as proposed by the authors. The methods are sound and well describe.

The one suggestion that I have is that it would be helpful to have some greater understanding of the variability in responses of different sampling locations to rainfall, within the stratified sampling types, i.e, among similar sites that one would expect to be the same. This may be particularly important for the grass response variability, and the Forbes' responses on sandy soils.

Following from this, there is no presentation of the spatial variation in responses. It would be great to see the kinds of spatial scale that the demonstrated responses manifest. This could be a simple map of deviation from predicted mean value for a site based on the stats analyses. Not only the absolute values to see how water related responses manifest, but also the deviation from expectation, even if there is not a clear explanation for such deviation.

Validity of the findings

Although the study is from one study site, the work is robust, and applicable across similar soil/vegetation/rainfall situations. The sample sizes ar adequate, the analyses sound, and properly conducted. The conclusions are clearly stated, and relate to the origins all research questions. The only suggestion I have is to probe more the spatial variation to complement the temporal analyses.

Additional comments

This may appear on first impression a relatively trivial study, in that we know that plants response to rainfall. However, the way that the authors pose the questions, and conduct the sampling and analyses, make this an important contribution to mechanistic understanding, which can be used as a basis to explain heterogeneity in animal behavior responses by elephant and other species. Such understanding is increasingly necessary as space for large herbivores is limited, and populations in restricted reserves are increasing. The authors may want to emphasis this aspect a bit more strongly in the discussion, and also link better to our increasing knowledge of mechanistic understanding of elephant decision making, such as the nutrient base decisions in the papers by Pretorius et al, Khohi et al, and Woolley et al., but also, for example, how stress affects movement strategies as per Jachowski et al paper in functional ecology.

---

## Round 0.2 · accepted · Accept

I enjoyed the MS and learned something worthwhile.